# NF-κB Decoy ODN-Loaded Poly(Lactic-co-glycolic Acid) Nanospheres Inhibit Alveolar Ridge Resorption

**DOI:** 10.3390/ijms24043699

**Published:** 2023-02-12

**Authors:** Albert chun-shuo Huang, Yuji Ishida, Kai Li, Duantawan Rintanalert, Kasumi Hatano-sato, Shuji Oishi, Jun Hosomichi, Risa Usumi-fujita, Hiroyuki Yamaguchi, Hiroyuki Tsujimoto, Aiko Sasai, Ayaka Ochi, Hajime Watanabe, Takashi Ono

**Affiliations:** 1Department of Orthodontic Science, Graduate School of Medical and Dental Sciences, Tokyo Medical and Dental University (TMDU), Tokyo 113-8549, Japan; 2Department of Orthodontics, Faculty of Dentistry, Chulalongkorn University, Bangkok 10330, Thailand; 3Department of Pediatrics, McGovern Medical School, The University of Texas Health Science Center at Houston, Houston, TX 77030, USA; 4Pharmaceutical/Beauty Science Research Center, Material Business Division, Hosokawa Micron Corporation, Osaka 573-1132, Japan; 5AnGes Inc., Tokyo 108-0014, Japan

**Keywords:** nuclear factor-kappa B, oligodeoxynucleotide, NF-κB transcription factor decoy, poly(lactic-co-glycolic acid) copolymer, nanosphere, tooth extraction, alveolar bone loss, bone remodeling, inflammation, wound healing

## Abstract

Residual ridge resorption combined with dimensional loss resulting from tooth extraction has a prolonged correlation with early excessive inflammation. Nuclear factor-kappa B (NF-κB) decoy oligodeoxynucleotides (ODNs) are double-stranded DNA sequences capable of downregulating the expression of downstream genes of the NF-κB pathway, which is recognized for regulating prototypical proinflammatory signals, physiological bone metabolism, pathologic bone destruction, and bone regeneration. The aim of this study was to investigate the therapeutic effect of NF-κB decoy ODNs on the extraction sockets of Wistar/ST rats when delivered by poly(lactic-co-glycolic acid) (PLGA) nanospheres. Microcomputed tomography and trabecular bone analysis following treatment with NF-κB decoy ODN-loaded PLGA nanospheres (PLGA-NfDs) demonstrated inhibition of vertical alveolar bone loss with increased bone volume, smoother trabecular bone surface, thicker trabecular bone, larger trabecular number and separation, and fewer bone porosities. Histomorphometric and reverse transcription–quantitative polymerase chain reaction analysis revealed reduced tartrate-resistant acid phosphatase-expressing osteoclasts, interleukin-1β, tumor necrosis factor-α, receptor activator of NF-κB ligand, turnover rate, and increased transforming growth factor-β1 immunopositive reactions and relative gene expression. These data demonstrate that local NF-κB decoy ODN transfection via PLGA-NfD can be used to effectively suppress inflammation in a tooth-extraction socket during the healing process, with the potential to accelerate new bone formation.

## 1. Introduction

Tooth extraction followed by early excessive inflammation often leads to progressive atrophy of the residual ridge, which causes alveolar bone deformities [1]. Bone resorption involves two phases: a drastic vertical reduction caused by bundle bone resorption, followed by overall horizontal and vertical tissue contraction including resorption of the outer surfaces of bone walls; resorption gradually occurs throughout life [2]. Although previous studies [3,4] have summarized the changes in bone physiology, the mechanism underlying the short-term and remodeling stages following tooth extraction, which in turn cause residual ridge resorption (RRR), remains unknown. Such dimensional loss can negatively affect the possibilities of clinical alternatives for subsequent restorative dental therapy [5,6].

Numerous approaches were previously explored to modulate inhibitors of the nuclear factor-kappa B (NF-κB) signaling pathway, which is widely recognized for regulating prototypical proinflammatory signals, physiological bone metabolism, pathologic bone destruction, and bone regeneration [7]. Decoy oligodeoxynucleotides (ODNs) are double-stranded DNA fragments that possess the same sequence as binding sites of specific transcription factors [8]. In particular, NF-κB decoy ODNs are capable of downregulating the expression of downstream genes of the NF-κB pathway, such as proinflammatory cytokine and osteoclastogenesis genes [9]. Indeed, inhibition of NF-κB using decoy ODNs was reported to be effective against in vitro osteoclast differentiation and activation, as well as to effectively prevent bone resorption in vivo [10]. Hence, decoy ODNs may hold therapeutic value for various bone metabolic diseases. However, the application of NF-κB decoy ODNs to mediate alveolar bone extraction socket healing remains poorly explored.

Poly(lactic-co-glycolic acid) (PLGA) synthesized as nanospheres has been used as an efficient vector for drug delivery for decoy ODNs in the nuclear medical field owing to its safety, enhanced stability, bioavailability, and long-term release properties [11]. Moreover, the improved pharmacokinetics of PLGA nanospheres loaded with NF-κB decoy ODNs suggests that it may represent a promising strategy to effectively inhibit the NF-κB transcriptional activity in the inflammatory process; NF-κB decoy ODN-loaded PLGA nanospheres (PLGA-NfD) were reported to exhibit excellent affinity for and adsorption to the surfaces of anionic cells derived from phosphate groups when introduced into cells through the mechanism of receptor-mediated endocytosis [12,13]. Nonetheless, the use of PLGA-NfD on tooth-extraction socket healing has yet to be investigated in vivo.

This study aimed to assess the therapeutic potential of NF-κB decoy ODN delivered by PLGA nanospheres for the prevention of bone loss in extraction sockets caused by early inflammation. Thus, we investigated the downstream effects of NF-κB inhibition on the expression of proinflammatory cytokines and osteoclastogenesis genes in rat extraction socket tissues during the early healing stage. Moreover, we surveyed the possibility of persistent long-term effects of decoy ODN-loaded PLGA nanospheres induced by local administration on alveolar bone tissues in vivo.

## 2. Results

### 2.1. PLGA-NfD Prevents Vertical Bone Loss of the Tooth-Extraction Socket and Preserves the Alveolar Ridge

Microcomputed tomography analysis demonstrated that the vertical bone height of extraction sockets decreased in all experimental groups on days 7 and 28 post tooth extraction (D7 and D28; Figure 1A–C). PLGA-NfD-treated rats showed a significantly decreased height loss in the buccal and middle aspects compared with the phosphate-buffered saline (PBS), naked scrambled decoy ODN (ScD), and PLGA-scrambled decoy ODN (PLGA-ScD) groups on D7, with a significant decrease in the palatal and distal aspects compared with the PBS and ScD groups (Appendix A). On D7, naked NF-κB decoy ODN (NfD)-treated rats showed a decreasing tendency of the buccal aspect compared with ScD-treated rats, but no significant difference was observed when compared with the PLGA-NfD-treated group. No significant differences were observed between the control and experimental groups in the mesial aspect on D7. Moreover, on D28, the PLGA-NfD-treated group showed significantly decreased height loss in all aspects compared with the PBS-treated group, significantly decreased height loss compared with the ScD-treated group in the buccal and distal aspects, decreased loss compared with the NfD-treated group in the palatal, mesial, and distal aspects, and a decreased loss compared with the PLGA-ScD-treated group in the palatal aspect.

Representative 3D images of volume of interest (VOI) of the extraction socket in all groups are shown in Figure 1D. In the PLGA-NfD-treated rats, the bone volume fraction (BV/TV, %), bone mineral density (BMD, mg/cm^3^), trabecular thickness (Tb.Th, µm), trabecular number (Tb.N, per mm), and trabecular star volume (V*tr, mm^3^) were significantly increased on D7 compared with PBS-, ScD-, and PLGA-ScD-treated rats (Figure 1E–M; Appendix A). Treatment with NfD also significantly increased the BMD and Tb.Th of the bone in the tooth-extraction socket compared with the ScD-treated rats and the NfD-, ScD-, and PBS-treated groups, respectively. In contrast, the bone surface ratio (BS/BV, per mm), trabecular separation (Tb.Sp, µm), trabecular spacing (Tb.Spac, µm), and bone marrow space star volume (V*m.space, mm^3^) upon PLGA-NfD treatment were significantly more decreased on D7 than on PBS-, ScD-, and PLGA-ScD-treated rats on the same day. NfD-treated rats had significantly reduced BS/BV and V*m.space compared with ScD- and PBS-treated rats, respectively. On D28, a similar tendency was observed in the PLGA-NfD-treated group, with significantly increased BV/TV, BMD, Tb.Th, Tb.N, and V*tr, but decreased BS/BV, Tb.Sp, Tb.Spac, and V*m.space, than in the control group. In contrast, no significant difference was observed between rats administrated NfD and PBS, which showed similar outcomes concerning the different features evaluated.

### 2.2. PLGA-NfD Prevents Early Inflammation and Promotes Tissue Remodeling in the Tooth-Extraction Socket

On D7, hematoxylin and eosin (H&E) staining revealed that inflammatory infiltrates were reduced, whereas the amount of woven and trabecular bone present in PLGA-NfD-treated rats was increased. Moreover, the activity of tartrate-resistant acid phosphatase (TRAP) staining was reduced, whereas that of alkaline phosphatase (ALP) was enhanced (Figure 2A–C). On D28, the PBS-, ScD-, NfD-, and PLGA-ScD-treated animals were characterized by relatively increased inflammatory infiltrates and decreased trabecular bone presentation. Immature bone formation was also relatively increased in these groups compared with samples from PLGA-NfD-treated rats. In contrast, PLGA-NfD treatment promoted a relatively higher and organized degree of bone formation, with infiltration of fewer inflammatory cells, associated with thicker bone trabeculae. TRAP and ALP staining results showed a similar tendency to those observed on D7.

Semi-quantitative analysis of the number of TRAP-positive osteoclasts on D7 showed that they were significantly decreased in the PLGA-NfD-rats group compared with the other groups (Figure 2D; Appendix A). Furthermore, NfD-treated rats showed a significantly decreased number of TRAP-positive osteoclasts than those treated with PBS, ScD, and PLGA-ScD, but significantly increased when compared with the PLGA-NfD group. On D28, all groups showed more moderate TRAP expression than on D7. The PLGA-NfD group showed significantly lower values than the other four groups. A contradictory tendency was observed concerning ALP activity, with the PLGA-NfD group showing significantly increased ALP-positive tissues on D7 and D28 than the other groups (Figure 2E; Appendix A).

### 2.3. PLGA-NfD Promotes Bone Formation in the Tooth-Extraction Socket

Triple-fluorescence bone labeling with calcein (green), demeclocycline hydrochloride (yellow), and alizarin complexone (red) was performed on D28 in cross sections of extraction sockets of all treatment groups. Overall, the calcein-to-demeclocycline-labeled surface showed a larger distance than the demeclocycline-to-alizarin-labeled surface. Notably, samples of PLGA-NfD-treated rats showed a tendency for increased bone formation in both calcein-to-demeclocycline and demeclocycline-to-alizarin-labeled surfaces compared with the other groups (Figure 3).

### 2.4. PLGA-NfD Prevents Early Inflammatory Activity and Its Negative Effects in the Tooth-Extraction Socket

Immuno-histomorphometric analyses showed positive staining for interleukin (IL)-1β, and tumor necrosis factor (TNF)-α was mainly observed in inflammatory infiltrates in the intramedullary area of the newly formed trabecular bone (Figure 4A,B). Positive staining for transforming growth factor (TGF)-β1 was observed in the endothelial and fibroblast-like cells, and positive staining for receptor activator of nuclear factor-kappa B ligand (RANKL) was observed in the osteoblastic lining cells closer to the alveolar bone (Figure 4C,D). On D7 and D28, the inflammatory reaction was reduced in PLGA-NfD-treated rats, which was demonstrated by a decrease in the ratio of IL-1β and TNF-α levels (Appendix A). Decreased bone resorption was observed by reduced RANKL expression. In contrast, promotion of bone formation was also observed with increased expression of TGF-β1. In NfD-treated rats, significantly decreased expression of IL-1β and RANKL was observed on D7, whereas the expression of the other evaluated molecules was similar to that in PBS, PLGA-ScD, and PLGA-NfD groups on D7 and D28 (Figure 4E–H).

Biochemical analysis of the relative gene expression of IL-1β and TNF-α confirmed that treatment with PLGA-NfD prevented early inflammation, with reduced levels of *Il1b* and *Tnf* on D7 compared with the PBS, ScD, and PLGA-ScD groups (Figure 5A–F, Appendix A). On D28, PLGA-NfD caused a significant decrease in *Il1b* expression compared with PBS alone, whereas the same significant difference was also found in *Tnf* expression upon treatment with ScD. Regarding osteoclastic-activity-related genes, *Tnfsf11* and *Tnfrsf11b* were found to be significantly decreased in PLGA-NfD-treated rats than on D7 upon PBS, ScD, and PLGA-ScD treatment. On D28, PLGA-NfD-treated animals showed significantly decreased *Tnfsf11* expression compared with PBS alone, whereas the same significant difference was also observed in *Tnfrsf11b* expression upon ScD treatment. RANKL/OPG (osteoprotegerin) (*Tnfsf11*/*Tnfrsf11b*) ratio in the PLGA-NfD group was significantly decreased compared with that in the PLGA-ScD group on D7; however, no significant difference was observed on D28. Regarding osteogenesis-related genes, the relative *Tgfb1* expression was increased in PLGA-NfD rats on both D7 and D28 compared with that in ScD-treated rats. In contrast, on D28, a significantly increased expression of *Tgfb1* was also observed upon PLGA-NfD treatment compared with PBS alone.

## 3. Discussion

To the best of our knowledge, this is the first in vivo study to demonstrate that PLGA-NfDs can prevent excessive inflammation and enhance alveolar healing after tooth extraction. Despite several studies on RRR [14,15], its etiological mechanism remains unclear. Nonetheless, the healing process of a tooth-extraction socket is usually initiated by an inflammatory phase as the beginning of a physiological immunological process of defense toward trauma, which inevitably cannot be prevented [3,16]. Moreover, the continuity of this sequential aftereffect can generate a considerably greater risk of alveolar ridge dimensional loss in the bone remodeling stage.

Treatment with ScD alone was one of the negative controls used in the current study; a similar experimental design has also been implemented in previous studies [8,10,17,18]. Herein, treatment with ScD and PLGA-ScD exhibited results identical to the vehicle control (PBS) in all analyses. Although previous research reported a therapeutic effect of PLGA in the extraction socket concerning the alveolar bone height, in the present study, the NfD group shared identical tendencies with the other groups (except PLGA-NfD) in almost all D7 analyses. On D28, the NfD treatment showed diminished effects, similar to that of the other groups except for PLGA-NfD treatment. These results indicate the necessity of PLGA as a vector to achieve a therapeutic effect, which indirectly proves the distinctive effects of PLGA loaded with NfD. According to previous meta-analyses, tooth extraction always triggers a process of bone resorption [5,19,20], in which the alveolar ridge undergoes progressive atrophy that is more noticeable in the apico-coronal dimension. The present findings demonstrate that PLGA-NfD administration can facilitate the maintenance of alveolar bone height. Furthermore, PLGA-NfD treatment can promote sustainable effects, as indicated by positive changes in the trabecular bone parameters following tooth loss. Morphological findings determined by microcomputed tomography demonstrated the preservative effects of PLGA-NfD, with inhibition of bone resorption not only in short term but also in remodeling periods.

While describing the histological process of socket healing, previous studies reported that numerous osteoclasts were present on the outer surface of the crest, with prominent osteoclastic activity resulting in resorption of both the buccal and palatal bone walls [4,21]. In the current study, all groups showed similar tendencies. On D7, all groups showed the histologically apparent beginning of the healing process with newly formed trabecular bone, whereas the PLGA-NfD group demonstrated reduced proliferative inflammatory infiltrates and TRAP activity along with increased ALP activity. Therefore, PLGA-NfD holds potential for preventing early alveolar bone resorption and promoting bone formation in the extraction socket. On D28, mature and well-defined bony trabeculae filled a large portion of the alveolar socket with multiple little islands of bone marrow and connective tissue. Although evidence of a reduced inflammatory reaction was noted on D28 among all groups compared with that on D7, less inflammatory infiltrate in the intramedullary area of the bone marrow among sections was observed in PLGA-NfD samples, revealing reduced bone resorption. These histological phenomena further support the hypothesized ability of PLGA-NfD to inhibit excessive inflammation with persistent effects sustained even up to the late healing phase of the extraction socket. Although increased ALP activity was detected within the structural components of the bone matrix of PLGA-NfD rats, a persistent positive effect toward bone formation during the remodeling stage of healing was observed through in vivo dynamic bone labeling. Indeed, a tendency of increased distance in both calcein-to-demeclocycline- and demeclocycline-to-alizarin-labeled surfaces revealed that the mineralization of newly formed bone also took place in PLGA-NfD-treated rats until D28. These results indicate that PLGA-NfD not only inhibited bone resorption, but also showed potential for bone healing paralleling with the short-term and remodeling phases.

Pathogenic stimulus, such as those from bacterial infections, in the beginning of and during socket healing are one of the most common reasons for early excessive inflammation and alveolar bone loss driven by immune response apart from traumatic inflammation [22]. This local bone loss was reported to be partly mediated by inflammatory infiltrates, including neutrophils, lymphocytes, plasma cells, and macrophages, which subsequently regulate the balance and survival of osteoclasts and osteoblasts (Figure 6). In the present study, increased levels of IL-1β and TNF-α in the PBS, ScD, and PLGA-ScD groups illustrated that intervention with these solutions did not suppress the normal physiological healing process, which begins with the inflammatory phase. Previous studies defined the basic multicellular unit (BMU) as a balance between bone resorption and formation, including osteoclasts and osteoblasts [23,24,25]. Based on the pharmacological mechanism of NfD, it can be inferred that the balance of bone resorption and formation might have been altered because of the decoy effect of NF-κB during the healing process of the alveolar socket. A previous in vitro study showed the selective NfD uptake into monocytes/macrophages, leaving other cells, such as the stromal and osteoblast cells, unaltered. Hence, the effect of NfD was entirely confined to osteoclasts and their progenitor cells, causing reduced migration of osteoclast precursor cells [26,27]. Based on previous correlated in vitro research and our in vivo immunohistochemical and biochemical results, the therapeutic mechanism of NfD may be explained by the downregulation of the expression and secretion of proinflammatory cytokines (IL-β and TNF-α) in and by inflammatory cells, which resulted in reduced stimulation toward the differentiation of osteoclast precursors and consequently reduced activation of mono- and multi-nucleated osteoclasts and polarized resorbing osteoclasts. In addition, because of the intracellular uptake by endocytosis of PLGA-NfD, downregulation of the expression of downstream osteoclastogenic genes, such as *NFATC1* and *TRAP*, in osteoclast precursors may also be triggered by PLGA-NfD, which directly hinders their differentiation and consequently causes a significant reduction in the number of these cells, also affecting normal stromal cells, osteoblasts, and osteocytes and reducing osteoblastic RANKL expression (Figure 6). Consequently, the decrease in RANKL production in the BMU can generate an environment of attenuated inflammation, causing the inhibition of RANKL activation and thereby preventing bone resorption. Interestingly, we found that the decoy ODN alone can downregulate the levels of inflammatory cytokines, which leads to reduced osteoclastic activity. In the current study, NfD-treated rats on D7 and PLGA-NfD-treated rats on D7 and D28 had increased levels of TGF-β1, which suggests an osteogenic tendency in socket healing. TGF-β1 is known as an immunoregulatory cytokine and bone-derived factor in osteoimmunology; however, when at a high concentration it can enhance osteoblast proliferation and downregulate RANKL expression in osteoblasts [28,29]. This also accounts for the reduced expression of RANKL in the present study. In other treatment groups that presented a lower TGF-β1 expression, osteoclast maturation was facilitated and, even though TGF-β1 expression increased via its normal physiological mechanism, the potential of bone formation still could not reach the same level as bone resorption on D28.

Because of the important role of NF-κB in the differentiation and activation of osteoclasts, its selective inhibition has been explored in previous studies as a way to block osteoclastogenesis [30,31]. Among a variety of approaches, which include mere temporary downregulation and gradual reduction by different pathways under transcription factor activity [32], decoy ODNs are a relatively sharper approach owing to their capability of reducing existing transcription factor activity and efficiently suppressing gene expression when one or more transcription factors interact with a single, related cis-element or when those factors are constitutively produced [33]. However, one of the major limitations of this approach is the rapid degradation of phosphodiester ODNs by intracellular nucleases; this further emphasizes the importance of PLGA as vector [34,35]. Previous studies concluded that PLGA nanospheres are a promising delivery system for double-stranded decoy ODNs targeting NF-κB [36,37]. Such a system allows sustained ODN release along with inhibition of the transcriptional activity of NF-κB in activated macrophages at significantly lower concentrations than naked ODN [38]. Other in vivo studies also reported its biocompatibility and biodegradability as properties of a potential and promising carrier for oral delivery [39,40]. A previous study reported on the characteristics of technical sensitivity and the time-consuming nature of periodontal regenerative surgery in clinical dentistry [41]. However, as a less invasive, safe, and more manipulative means for topical administration of PLGA [42], PLGA-NfD can be considered an innovative alternative to periodontal regenerative surgery for future clinical utilization.

## 4. Materials and Methods

### 4.1. Preparation of Decoy ODN Nuclear Medicine

Naked scrambled decoy ODN, also known as phosphorothioated double-stranded scrambled decoy ODN (with the sequences 5ʹ-TTGCCGTACCTGACTTAGCC-3ʹ and 3ʹ-AACGGCATGGACTGAATCGG-5ʹ), and naked NF-κB decoy ODN, also known as phosphorothioated double-stranded NF-κB decoy ODN (with sequences 5ʹ-CCTTGAAGGGATTTCCCTCC-3ʹ and 3ʹ-GGAACTTCCCTAAAGGGAGG-5ʹ), were used. The concentrations of scrambled decoy ODN in the ScD and PLGA-SCD solutions, as well as the NF-κB decoy ODN in the NfD and PLGA-NfD solutions were 0.02% *w*/*v* (0.2 mg/mL). The concentration of hydroxypropyl cellulose-H used was 3.3% (*w*/*v*) with 2.0% PLGA nanospheres (20 mg/mL). Research regents relating to the NF-κB decoy ODN and PLGA-NfD were provided by AnGes Inc. (Osaka, Japan) and Hosokawa Micron Corporation (Osaka, Japan).

### 4.2. Physicochemical Properties of Decoy ODN-Loaded PLGA Nanosphere

The particle size of PLGA-ScD and PLGA-NfD nanospheres was measured by dynamic light scattering (DLS) method (Table 1). Moreover, the content percentage (%) of the ODNs in PLGA was measured by UV spectrophotometry in the current research. The encapsulation efficiency of ODN in PLGA was calculated using the following formula: [active pharmaceutical ingredients (API)/API + PLGA + polyvinyl alcohol (PVA)] (Figure 7, Table 1).

### 4.3. Experimental Animals

A total of 62 Wistar/ST male rats (6 weeks old) (Sankyo Lab Service, Tokyo, Japan) were used in compliance with the in vivo experiment (ARRIVE) 2.0 guidelines. All animals were housed in the same room with controlled temperature, humidity, and light. A standard alternating 12 h light/dark cycle was maintained. The health status and body weight of the rats were monitored every other day.

### 4.4. Surgical Procedure of Teeth Extraction

The rats were randomly divided into 5 treatment groups, each containing 12 animals. The treatment groups were: vehicle control (PBS), naked scrambled decoy ODN (ScD), naked NF-κB decoy ODN (NfD), PLGA-scrambled decoy ODN (PLGA-ScD), and PLGA-NF-κB decoy ODN (PLGA-NfD) (Appendix A). All rats underwent bilateral maxillary first molar extraction surgery under general anesthesia, conducted by subcutaneous injection of a mixed anesthetic (medetomidine (Nippon Zenyaku Kogyo Co., Ltd., Fukushima, Japan), 0.3 mg/kg; midazolam (Teva Takeda Yakuhin Ltd., Nagoya, Japan), 4 mg/kg; butorphanol (Meiji Seika Pharma Co., Ltd., Tokyo, Japan), 5 mg/kg), followed by bilateral maxillary first molars extracted by specific forceps. Immediately after the extraction, 0.9% PBS (pH 7.4) (Nacalai Tesque, Inc., Kyoto, Japan) and the specific nuclear medicines listed above were locally administered by intraosseous injection into the bilateral extraction socket according to the group design (0.25 mL per extraction socket region). No postoperative complications or syndromes were observed in any animal.

### 4.5. Dynamic Fluorescent Labeling of the Extraction Socket

For in vivo dynamic fluorescent labeling of the extraction sockets, two animals from each group were administered calcein (20 mg/kg; Sigma-Aldrich, St. Louis, MO, USA), demeclocycline hydrochloride (20 mg/kg; Sigma-Aldrich, St. Louis, MO, USA), and Alizarin complexone (20 mg/kg; ALC, Dojindo, Kumamoto, Japan) subcutaneously on days 6, 15, and 24 after tooth extraction, respectively. The samples were then thoroughly washed with PBS before fixation using 10% PBS-based formaldehyde fixative (pH 7.4) (Fujifilm Wako Pure Chemical Corp., Osaka, Japan) for 48 h at 4 °C under constant shaking motion. The undecalcified frozen blocks were prepared using the same method into 5-μm thick tissue sections retrieved via adhesive Kawamoto film (Section-Lab Co., Ltd., Hiroshima, Japan). Bone formation on the extraction sockets according to the bone-labeling schedule was observed using a BZ-X700 fluorescence microscope (Keyence Corp., Osaka, Japan).

### 4.6. Tissue Preparation

A split-mouth design was prepared for maxillary right extraction socket tissue for alveolar bone morphological and histomorphometric evaluation (*n* = 5) and maxillary left extraction socket tissue for biochemical evaluation (*n* = 5). After 7 and 28 days from teeth extraction, 5 animals from each group were euthanized using carbon dioxide gas. Maxillae with tooth-extraction socket and the surrounding tissue were immediately collected. For morphological and histomorphometric samples, the right hemimaxilla with extraction socket was fixed with 4% paraformaldehyde (pH 7.4) (Fujifilm Wako Pure Chemical Corp., Osaka, Japan) for 48 h at 4 °C. For biochemical evaluation, the left hemimaxilla, including extraction socket tissues, was resected. Tissue samples of the extraction socket were transferred into liquid nitrogen immediately after collection.

### 4.7. Morphological Evaluation

#### 3D Microcomputed Tomography Analysis

Alveolar bone morphological evaluation of the extraction sockets was performed using ex vivo 3D microcomputed tomography. Tissue samples were scanned using InspeXio SMX-100CT computed tomography system (Shimadzu Corp., Kyoto, Japan) and analyzed using a 3D trabecular bone analysis software (TRI/3D-BON-FCS; RATOC System Engineering Co., Ltd., Tokyo, Japan) according to the manufacturer’s instructions. All fixed tissue samples were scanned with output settings of 75 kV and 140 mA, and a scanning resolution of 8.0 μm. The VOI for the 3D microstructural morphometry analysis was defined by the borders, including the total tooth-extraction socket region with a grid area of 25.3 mm^3^ (LX: 2.5 mm, LY: 2.2 mm, and LZ: 4.6 mm). Calibration and adjustment were performed by the reference line of the mid-palatal suture plane and the maxillary palatal transverse plane of each sample (Figure 8). Vertical height loss of the extraction socket was measured and defined by vertical bone height changes of the extraction socket on D7 and D28 separately. The cement–enamel junction of M3 to the alveolar bone crest (ABC) of the M1 extraction socket was determined as the height changes of the extraction socket. The buccal, middle, palatal, mesial, and distal aspects enclosing the extraction socket area were measured and evaluated. Trabecular bone analysis was performed using the selected VOI, identified by the direct-measures technique [43]. Trabecular bone was evaluated using the following parameters: bone volume fraction (BV/TV, %), bone mineral density (BMD, mg/cm^3^), bone surface ratio (BS/BV, per mm), trabecular thickness (Tb.Th, µm), trabecular number (Tb.N, per mm), trabecular separation (Tb.Sp, µm), trabecular spacing (Tb.Spac, µm), bone marrow space star volume (V*m.space, mm^3^), and trabecular star volume (V*tr, mm^3^).

### 4.8. Histomorphometric Evaluation

After microcomputed tomography analysis, the specimens were decalcified with 10% disodium ethylenediamine tetraacetate (pH 7.4) (Nacalai Tesque, Inc., Kyoto, Japan) at 4 °C for 6 weeks and were embedded in paraffin (Leica Biosystems, Nussloch, Germany) through standard dehydration and paraffin infiltration steps. The paraffin-embedded tissues were cut at 4-μm thickness using a rotary microtome (Leica Biosystems, Nussloch, Germany) parallel to the sagittal plane of the right hemimaxilla. Histomorphometric evaluations included the histological observations of stained tissue sections examined under DXm1200 light microscopy (Nikon, Kanagawa, Japan) using the NIS-Elements D Imaging Software (Version 2.30; Nikon, Kanagawa, Japan). The images were analyzed using ImageJ software (version 1.52; National Institutes of Health, Bethesda, MD, USA). The region of interest was determined to be a 330 μm × 409 μm region in the mesial root socket, which was considered representative of the extraction socket area. Analysis was performed after obtaining three randomized tissue sections for each sample with five random images at 200× magnification.

#### 4.8.1. Histochemical Staining of TRAP and ALP

To further analyze the catabolic activity in the alveolar bone, mononucleated and multinucleated osteoclasts, as well as polarized resorbing osteoclasts were detected by TRAP staining. The TRAP staining kit (Fujifilm Wako Pure Chemical Corp., Osaka, Japan) was used according to the manufacturer’s protocol. The number of TRAP-positive cells per one section and per mm^2^ of the region of interest were counted by a single examiner and the averages were calculated.

To assess bone formation in the extraction socket, ALP-positive stained area (%) was analyzed using an ALP staining kit (Fujifilm Wako Pure Chemical Corp., Osaka, Japan) at 37 °C for 30 min, according to the manufacturer’s instructions.

#### 4.8.2. Immunohistochemical Staining of Inflammatory Cytokines (IL-1β, TNF-α), Osteoclastogenic, and Osteogenesis Markers (RANKL and TGF-β1)

Sections were stained using the following primary antibodies for immunohistochemical analyses: anti-interleukin (IL)-1β (dilution ratio: 1:400) (Bioss Inc., Woburn, MA, USA); anti-tumor necrosis factor (TNF)-α (dilution ratio: 1:400) (Bioss Inc., Woburn, MA, USA); anti-transforming growth factor (TGF)-β1 (dilution ratio: 1:400) (Bioss Inc., Woburn, MA, USA); and anti-receptor activator of nuclear factor-kappa B ligand (RANKL) (dilution ratio: 1:400) (Bioss Inc., Woburn, MA, USA). After deparaffinization and rehydration, the samples were treated using 3% hydrogen peroxide (Abcam plc., Cambridge, U.K.) for 10 min to quench the endogenous peroxidase activity. After incubating with 30 min of normal goat serum to block non-specific binding at room temperature, primary antibodies with the specific concentrations listed above were added to the sections and incubated overnight at 4 °C. On the following day, VECTASTAIN Elite ABC Rabbit IgG Kit (Vector Laboratories, Inc.,, Burlingame, CA, USA) was used by incubating with a biotinylated secondary antibody for 30 min. Subsequently, the prepared VECTASTAIN ABC Reagent was applied to the slides, and sections were incubated for another 30 min. Sections were stained with 3,3ʹ-diaminobenzidine (DAB) (Abcam plc., Cambridge, U.K.) and counterstained with hematoxylin (Fujifilm Wako Pure Chemical Corp., Osaka, Japan). The protein expression levels of IL-1β, TNF-α, TGF-β1, and RANKL were semi-quantified by the percentage of immunopositive stained areas.

### 4.9. Relative Gene Expression Analysis of Inflammatory Cytokines and Osteoclastogenic/Osteogenesis Markers

The expression of genes related to inflammation and bone metabolism was examined by reverse transcription quantitative real-time PCR (RT-qPCR). Briefly, RNA was isolated from the alveolar bone socket as previously described [44]. Total RNA was isolated using Invitrogen TRIzol reagent (Thermo Fisher Scientific Inc., Waltham, MA, USA) and was reverse transcribed using PrimeScript RT Master Mix (Takara Bio Inc., Shiga, Japan) in accordance with the manufacturer’s instructions. Real-time PCR analysis was performed using the Probe qPCR Mix (Takara Bio Inc., Shiga, Japan) and Applied Biosystems 7500 Real-Time PCR System (Thermo Fisher Scientific Inc., Waltham, MA, USA). Appropriate specific Applied Biosystems TaqMan Gene Expression Assay primers (Thermo Fisher Scientific Inc., Waltham, MA, USA) were chosen for real-time PCR amplification of rat *Gapdh* (Rn01775763_g1), rat *Il1b* (Rn00580432_m1), rat *Tnf* (Rn01525859_g1), rat *Tnfsf11* (Rn00589289_m1), rat *Tnfrsf1b* (Rn00563499_m1), and rat *Tgfb1* (Rn00572010_m1). Relative gene levels were calculated using the comparative Ct method normalized to *Gapdh*. To assess the capability and degree of bone resorption and turnover of the extraction sockets, the RANKL/OPG ratio was evaluated, and relative gene expression of *Tnfsf11*/*Gapdh* over *Tnfrsf1b*/*Gapdh* was calculated.

### 4.10. Statistical Analysis

The normality was assessed using the Shapiro–Wilk test and the equality of variances was evaluated using Levene’s test. For parametric analysis, intergroup comparisons were performed via one-way analysis of variance followed by Tukey’s post hoc test for the analysis of microcomputed tomography-determined height loss and trabecular bone parameters, as well as for the analysis of TRAP, ALP, and immunohistochemical data (*n* = 5 for each group). Non-parametric Kruskal–Wallis test followed by the Dunn’s test for multiple comparisons were performed to assess differences in the relative gene expression analysis (*n* = 5 for each group). Statistical analysis was performed using IBM SPSS Statistics for Windows (version 27.0; IBM Corp., Armonk, NY, USA) and GraphPad Prism 9 (version 9.3.1; GraphPad Software Inc., San Diego, CA, USA). The results are presented as mean ± standard deviation (*n* = 5 per group). Statistical significance was accepted at *p* < 0.05.

## 5. Conclusions

In conclusion, this study demonstrated the importance of restoring the balance of the socket healing remodeling process that is disrupted by the acute early excessive inflammation caused by excessive osteoclastic activity, which results in net bone loss. By administering PLGA-NfD, the compromised BMU balance in the modeling–remodeling process can be restored and possibly manipulated to prevent the progression of early acute inflammation into long-term chronic inflammation in RRR and other alveolar bone-related syndromes.

## Figures and Tables

**Figure 1 ijms-24-03699-f001:**
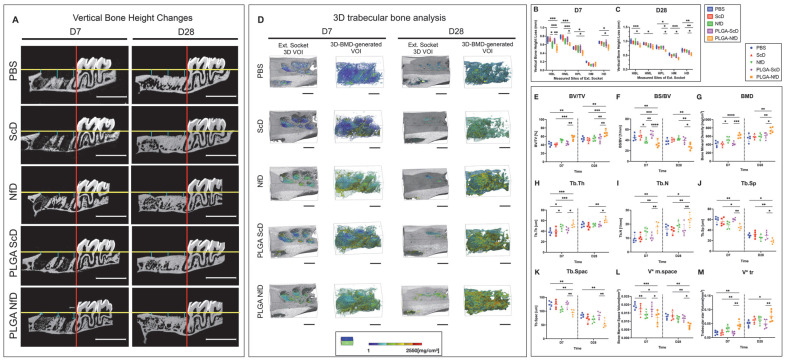
PLGA-NfD prevents vertical bone loss of the tooth-extraction socket and preserves the alveolar ridge. (**A**) Sagittal view of representative microcomputed tomography images. Blue lines indicate the extent of vertical alveolar bone loss in each group. (**B**,**C**) Linear measurement of the loss of vertical height was defined by vertical height changes of the socket. Measurements were performed in the VOI from the cement–enamel junction of M3 to the alveolar bone crest of the socket (scale bar = 3 mm). (**D**) Representative three-dimensional (3D) images of the VOI at the tooth-extraction socket. Trabecular 3D-BMD-generated VOI was used as the area of measurement for the alveolar bone analysis. The BMD value is indicated by the BMD color transition scale (scale bar = 1 mm). (**E**–**M**) 3D microcomputed tomography trabecular bone analysis of the tooth-extraction socket on D7 and D28. On D7, the PLGA-NfD and NfD groups showed an increase in BV/TV, BMD, Tb.Th, Tb.N, and V*tr and a decrease in BS/BV, Tb.Sp, Tb.Spac, and V*m.space; on D28, the PLGA-NfD group showed an increase in BV/TV, BMD, Tb.Th, Tb.N, and V*tr and a decrease in BS/BV, Tb.Sp, Tb.Spac, and V*m.space. Values are presented as mean ± standard deviation (*n* = 5). * *p* < 0.05, ** *p* < 0.01, *** *p* < 0.001, and **** *p* < 0.0001. Abbreviations: BMD, bone mineral density; BS/BV, bone surface ratio; BV/TV, bone volume fraction; D28, post-extraction day 28; D7, post-extraction day 7; HBL, buccal height loss of extraction socket; HD, distal height loss of extraction socket; HM, mesial height loss of extraction socket; HML, middle height loss of extraction socket; HPL, palatal height loss of extraction socket; NfD, naked NF-κB decoy; NF-κB, nuclear factor-kappa B; ODN, oligodeoxynucleotide; PBS, phosphate-buffered saline; PLGA, poly(lactic-co-glycolic acid); PLGA-NfD, NF-κB decoy ODN-loaded PLGA nanosphere; PLGA-ScD, scrambled decoy ODN-loaded PLGA nanosphere; ScD, naked scrambled decoy; Tb.N, trabecular number; Tb.Sp, trabecular separation; Tb.Spac, trabecular spacing; Tb.Th, trabecular thickness; V*m.space, bone marrow space star volume; V*tr, trabecular star volume; VOI, volume of interest.

**Figure 2 ijms-24-03699-f002:**
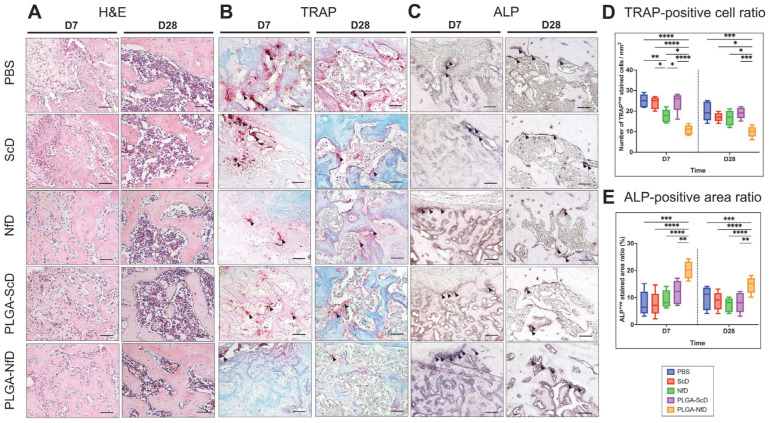
Representative findings of the mesial root socket on D7 and D28 after M1 extraction in all groups. (**A**) Hematoxylin and eosin (H&E) staining; (**B**) tartrate-resistant acid phosphatase (TRAP) staining; (**C**) alkaline phosphatase (ALP) staining. Magnification, 200×. Black arrow indicates positive region. Scale bar = 100 μm. (**D**,**E**) Relative semiquantitative analysis of TRAP and ALP staining of the M1 extraction socket. Abbreviations: D7, post-extraction day 7; D28, post-extraction day 28; NfD, naked NF-κB decoy; NF-κB, nuclear factor-kappa B; ODN, oligodeoxynucleotide; PBS, phosphate-buffered saline; PLGA, poly(lactic-co-glycolic acid); PLGA-NfD, NF-κB decoy ODN-loaded PLGA nanosphere; PLGA-ScD, scrambled decoy ODN-loaded PLGA nanosphere; ScD, naked scrambled decoy. Values are presented as mean ± standard deviation (*n* = 5). * *p* < 0.05, ** *p* < 0.01, *** *p* < 0.001, and **** *p* < 0.0001.

**Figure 3 ijms-24-03699-f003:**
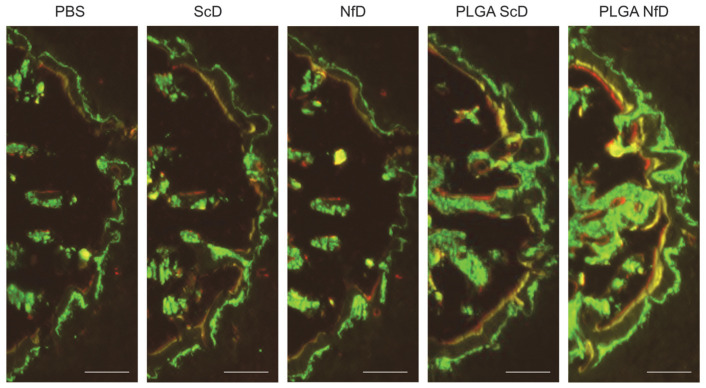
Assessment of dynamic fluorescent bone labeling of tooth-extraction sockets. Representative fluorescent images of the mesial portion of the M1 mesial socket labeled with fluorescent reagents targeting calcein (day 6), demeclocycline hydrochloride (day 15), and alizarin complexone (day 24) after tooth extraction. Magnification: 40×. Scale bar = 100 μm. Abbreviations: NfD, naked NF-κB decoy; NF-κB, nuclear factor-kappa B; ODN, oligodeoxynucleotide; PBS, phosphate-buffered saline; PLGA, poly(lactic-co-glycolic acid); PLGA-NfD, NF-κB decoy ODN-loaded PLGA nanosphere; PLGA-ScD, scrambled decoy ODN-loaded PLGA nanosphere; ScD, naked scrambled decoy.

**Figure 4 ijms-24-03699-f004:**
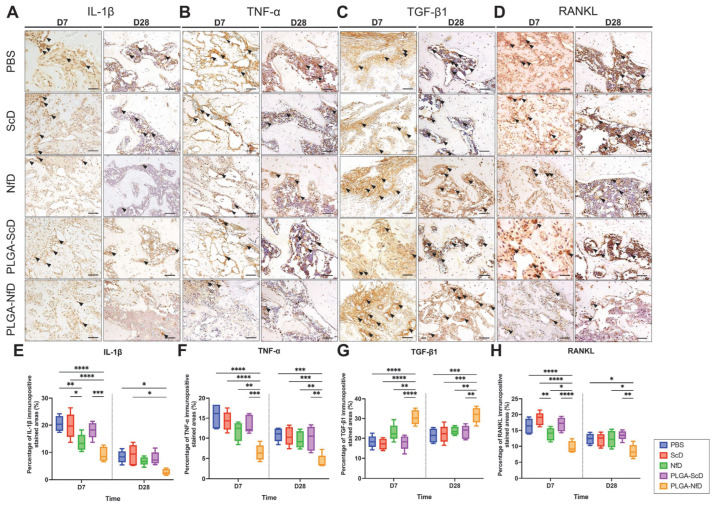
(**A**–**D**) Representative immunohistochemical staining of IL-1β, TNF-α, TGF-β1, and RANKL in the mesial root socket on D7 and D28 after M1 extraction. Magnification: 200×. Black arrows indicate immune-positive regions. Scale Bar = 100 μm. (**E**–**H**) Relative semi-quantitative analysis of immunopositive stained areas (%) of the extraction socket. Abbreviations: D7, post-extraction day 7; D28, post-extraction day 28; IL, interleukin; NfD, naked NF-κB decoy; NF-κB, nuclear factor-kappa B; ODN, oligodeoxynucleotide; PBS, phosphate-buffered saline; PLGA, poly(lactic-co-glycolic acid); PLGA-NfD, NF-κB decoy ODN-loaded PLGA nanosphere; PLGA-ScD, scrambled decoy ODN-loaded PLGA nanosphere; RANKL, receptor activator of nuclear factor-kappa B ligand; ScD, naked scrambled decoy; TGF, transforming growth factor; TNF, tumor necrosis factor. Values are presented as mean ± standard deviation (*n* = 5). * *p* < 0.05, ** *p* < 0.01, *** *p* < 0.001, and **** *p* < 0.0001.

**Figure 5 ijms-24-03699-f005:**
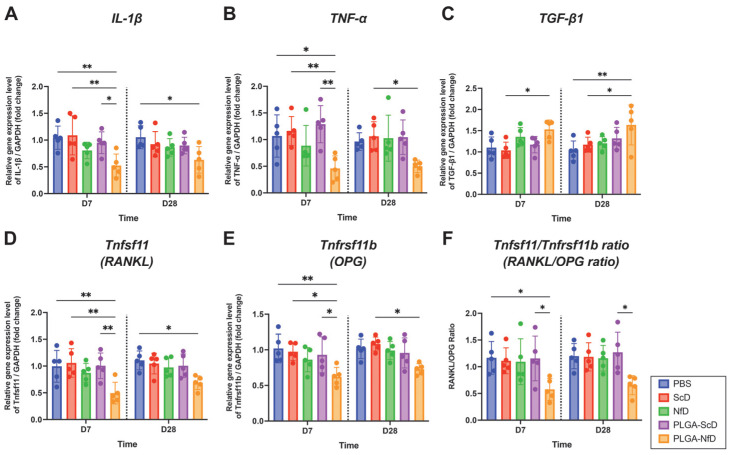
Quantitative real-time polymerase chain reaction analysis of inflammatory cytokines and osteoclastogenic/osteogenesis markers in the alveolar extraction bone tissue on D7 and D28 after tooth extraction. Relative gene expression of (**A**) *Il1b*, (**B**) *Tnf*, (**C**) *Tgfb1*, (**D**) *Tnfsf11* (RANKL), (**E**) *Tnfrsf11b* (OPG), and (**F**) *Tnfsf11/Tnfrsf11b* (RANKL/OPG) ratio were determined by reverse transcription quantitative real-time polymerase chain reaction analysis. *GAPDH* was used as the internal reference. Values are presented as mean ± standard deviation (*n* = 5). * *p* < 0.05, ** *p* < 0.01. Abbreviations: D7, post-extraction day 7; D28, post-extraction day 28; IL, interleukin; NfD, naked NF-κB decoy; NF-κB, nuclear factor-kappa B; ODN, oligodeoxynucleotide; OPG, osteoprotegerin; PBS, phosphate-buffered saline; PLGA, poly(lactic-co-glycolic acid); PLGA-NfD, NF-κB decoy ODN-loaded PLGA nanosphere; PLGA-ScD, scrambled decoy ODN-loaded PLGA nanosphere; RANKL, receptor activator of nuclear factor-kappa B ligand; ScD, naked scrambled decoy; TGF, transforming growth factor; TNF, tumor necrosis factor.

**Figure 6 ijms-24-03699-f006:**
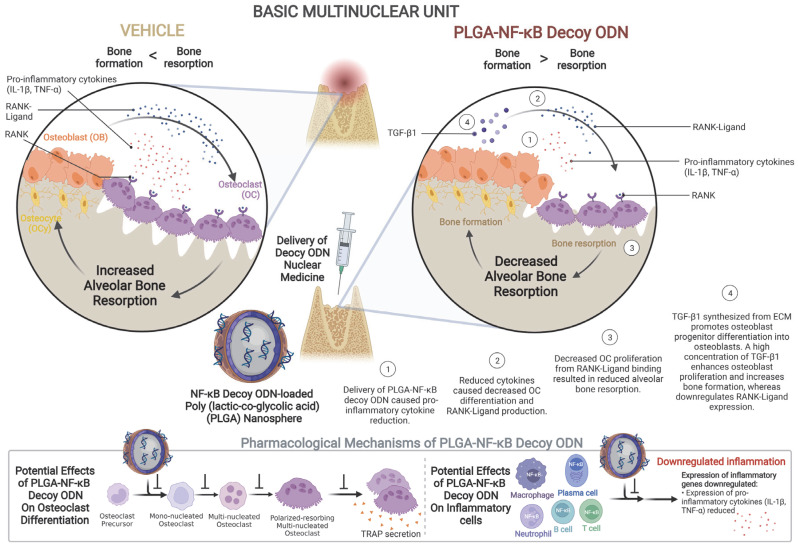
Schematic illustration of the effect of PLGA-NfD on osteoclast differentiation and inflammatory cells, including neutrophils, T- and B-lymphocytes, plasma cells, and macrophages on the extraction-socket tissue. PLGA-NfD demonstrates the usefulness of the PLGA nanospheres for NF-κB decoy ODN transfection into the extraction socket under inflammatory healing conditions. Local administration of PLGA-NfD has the clinical potential to prevent dimensional loss at the healing extraction socket and thereby allowing predictable prosthetic rehabilitation. Abbreviations: NF-κB, nuclear factor-kappa B; ODN, oligodeoxynucleotide; PLGA, poly(lactic-co-glycolic acid); PLGA-NfD, NF-κB decoy ODN-loaded PLGA nanosphere.

**Figure 7 ijms-24-03699-f007:**
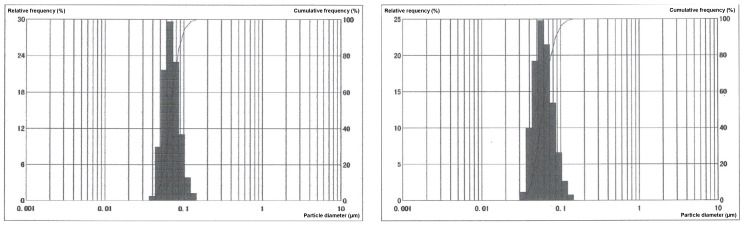
Particle size distribution of the PLGA-NfD (**left**) and PLGA-ScD (**right**) nanospheres evaluated by DLS method. The histogram indicates the relative frequency and the black line represents the cumulative distribution frequency as the particle count-basis.

**Figure 8 ijms-24-03699-f008:**
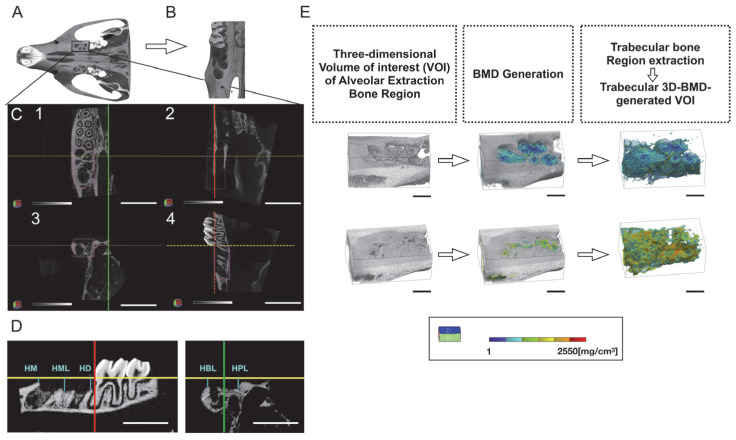
Assessment of 3D microcomputed tomography images of the maxillary extraction socket. Volume of interest (VOI) in the images of the extraction socket used for analysis is shown. (**A**,**B**) 3D microcomputed tomography scan of the extraction socket. (**C**) Reference plane: first, the mid-palatal suture plane of the maxillary left hemimaxilla in transverse view (C1, green line) was aligned, and then the hard palate referenced by the maxillary palatal plane of the sagittal view (C2, red line) and the coronal (C3, green line) planes were determined. The VOI was determined including the whole socket area by the border of the line of the M3 cement–enamel junction (C4, red dotted line) and the line of height of contour of M2 (C4, yellow dotted line). After the determination of VOI (C1-4), the VOI of the extraction socket was generated in a cuboidal volume (LX: 2.5 mm, LY: 2.2 mm, and LZ: 4.6 mm) (Scale bar = 4 mm). (**D**) Different aspects (left, parasagittal; right, frontal) of linear measurement of the buccal, middle, palatal, mesial, and distal enclosing the extraction socket area were evaluated as vertical height loss (scale bar = 1 mm). (**E**) Representative 3D images of the VOI of the extraction socket. Trabecular bone structures of the socket were extracted and reconstructed into a 3D image. After BMD generation, trabecular bone parameter measurements were conducted using the trabecular 3D-BMD-generated VOI. BMD value was reflected in the BMD color transition scale shown below (scale bar = 1 mm). Abbreviations: BMD, bone mineral density; HBL, buccal height loss of extraction socket; HD, distal height loss of extraction socket; HM, mesial height loss of extraction socket; HML, middle height loss of extraction socket; HPL, palatal height loss of extraction socket.

**Table 1 ijms-24-03699-t001:** Physicochemical properties of the decoy ODN-loaded PLGA nanospheres.

	Average Particle Size (µm)	Decoy ODN Content (%)
PLGA-ScD	0.070	1.03 wt%
PLGA-NfD	0.058	1.02 wt%

Abbreviations: NF-κB, nuclear factor-kappa B; ODN, oligodeoxynucleotide; PLGA, poly(lactic-co-glycolic acid); PLGA-NfD, NF-κB decoy ODN-loaded PLGA nanosphere; PLGA-ScD, scrambled decoy ODN-loaded PLGA nanosphere.

## Data Availability

Not applicable.

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
