# Peer review of "NF-κB Decoy ODN-Loaded Poly(Lactic-co-glycolic Acid) Nanospheres Inhibit Alveolar Ridge Resorption"

_ijms, 2023, doi:10.3390/ijms24043699_

Round 1

Reviewer 1 Report

This research demonstrates a relatively new area that has emerged from the lack of previous reports regarding the application of PLGA-NfD to the extraction socket; the scitific illutration is well organised and a lot of disscussion is included, overall, that is a good manuscript can be accept.

my only concern is that, no main figure data on the characterization of the  ODN-loaded PLGA nanospheres, it would be great if that part data can be demonstrated.

Reviewer 2 Report

The authors have proposed a manuscript named “NF-κB Decoy ODN-Loaded Poly(lactic-co-glycolic Acid) Nano-2 spheres Inhibit Alveolar Ridge Resorption”. They designed PLGA-NfD and tested its effects on the extraction socket. There are a few recommendations before further processing the manuscript.

1.      Shall the authors clarify the choice of ODN rather than siRNA or shRNA NF-Κb? Have you compared ODN with the siRNA strategy? How about the downregulation efficiency of ODN compared with siRNA?

2.      Characterizations such as electron microscopic analysis, DLS, zeta potential, and release kinetics of PLGA- NfD should be included.

3.      How about the encapsulation efficiency of ODN in PLGA? And how about the release profiles?

4.      Inhibiting NF-Κb activity may cause adverse consequences such as immune suppression and tissue damage. Did the PLGA-ODN used in this work have these concerns?

Round 2

Reviewer 2 Report

The paper is accepted.